# Lung Field Segmentation in Chest X-rays: A Deformation-Tolerant Procedure Based on the Approximation of Rib Cage Seed Points

**Vasileios Bosdelekidis [1] and Nikolaos S. Ioakeimidis [2,*]**

[1]   Department of Electrical and Computer Engineering, Polytechnic School,
     Aristotle University of Thessaloniki, 54124 Thessaloniki, Greece; vmposdel@auth.gr
[2]   Department of Medicine, School of Health Sciences, Aristotle University of Thessaloniki,
     54124 Thessaloniki, Greece
*   Correspondence: ioakeimn@auth.gr



**Featured Application:  The described simple and straightforward lung field segmentation algorithm could serve as a basis for automated Chest X-ray interpretation and aid the rapid discrimination of pathological Chest X-rays in high-volume radiology settings, remaining largely unaffected by variable lung shapes and chest deformities.**

**Abstract:** The delineation of bone structures is a crucial step in Chest X-ray image analysis. In the case of lung field segmentation, the main approach after the localization of bone structures is either their individual analysis or their suppression. We prove that a very fast and approximate identification of bone points that are most probably located inside the lung area can help in the segmentation of the lung fields, without the need for bone structure suppression. We introduce a deformation-tolerant region growing procedure. In a two-step approach, a sparse representation of the rib cage is guided to several support points on the lung border. We studied and dealt with the presence of other bone structures that interfere with the lung field. Our method demonstrated very robust behavior even with highly deformed lung appearances, and it achieved state-of-the-art performance in segmentations for the vast majority of evaluated CXR images. Our region growing approach based on the automatically detected rib cage points achieved an average Dice similarity score of 0.92 on the Montgomery County Chest X-ray dataset. We are confident that bone seed points can robustly mark a high-quality lung area while remaining unaffected by different lung shapes and abnormal structures.

**Keywords:** image segmentation; Chest X-ray (CXR); lungs; region growing; rib cage

---

## 1. Introduction

The Chest X-ray (CXR) is a common and widely used imaging modality in everyday clinical practice, aiming to visualize disorders of the bony thorax and the organs of the thoracic cavity [1,2]. Even though it is cheap, readily available and technically simple, it can reveal a significant amount of information regarding the patient's health, thus providing important clues for an accurate diagnosis [1]. Over the years, CXR has inevitably faced the rapid progress of medical imaging technology and has been subjected to criticism regarding its poor diagnostic sensitivity, which has to be counterbalanced by an accurate, detailed and time-consuming interpretation by a radiologist [1]. Radiograph interpretation is prone to error and discrepancies, and even an experienced radiologist using a systematic approach can miss 10–15% of pathologic lesions [1]. A discrepancy is defined as the extent of disagreement between the opinions of two radiologists regarding a diagnosis [3]. Studies report a day-to-day error rate of 3–5% per observer and a disagreement rate of 5–9% between two observers [3]. In a study

of 100 chest radiographs, concordance between three expert radiologists was 61%, pointing out the significant interobserver variation in CXR interpretation [4]. The structures and pathologies depicted on a CXR often have similar opacities or overlap one another, making the discrimination of fine lesions difficult. The abovementioned weaknesses paved the way for Computer-Aided Diagnosis (CAD) in chest radiography, aiming to improve the diagnostic accuracy. The first attempt to implement CAD in CXR interpretation dates back to 1963 when Lodwick et al. proposed a method to determine the probability of a lung nodule being cancerous, based on the systematic classification of unique descriptive features of lung lesions [5]. Since then, CAD in chest radiography has focused on automating discreet steps of CXR interpretation such as size and shape measurements, contour delineation and nodule detection [6]. Moreover, CAD research has focused on developing algorithms suited for one disease at a time (pneumonia, lung cancer lesions, tuberculosis, pneumothorax) and not an all-in-one automated solution [6]. Over the past few years, there has been an increasing number of published works aiming to identify various lung pathologies by using deep learning approaches such as Convolutional Neural Networks (CNN) [7–10].

Automated CXR interpretation could drastically alleviate the increasing workload of radiologists by complementing their diagnostic procedures and acting as a "second opinion" [11]. From a technical standpoint, a typical CXR analysis algorithm comprises the following discrete steps: defining the Region Of Interest (ROI), identifying specific imaging features of the ROI and using a machine learning strategy to classify the detected pathology [12]. In the case of CXR analysis, the ROI is the lung region within the bony thorax, and its correct identification is a crucial prerequisite. Accurately delineating the lung boundaries is a very challenging image analysis problem, and it is well studied in the literature. The majority of available algorithms are evaluated based on anatomically "correct" CXRs and are sometimes prone to error due to anatomical variations, chest deformities and incorrect positioning of the patient [12].

In the era of the ongoing pandemic of the novel coronavirus disease 2019 (COVID-19), which has pushed the critical capacity and efficiency of healthcare facilities to their limits, fast and accurate diagnosis of positive cases has become a matter of crucial importance. Hu et al. proposed a weakly supervised deep-learning method for the detection of COVID-19 from chest Computed Tomography (CT) scans [13]. The aim of their approach was to provide a solution for the fast detection of COVID-19, minimizing the time-consuming need for manual labeling of cases by radiologists. Their multiscale learning scheme was able to cope with variations in the size and location of the lesions, since pathological lesions are usually small for mild cases. They evaluated their algorithm on a chest CT dataset from multiple centers and CT scanners. Discrepancies in image appearance due to the use of different classes of scanners or variable image acquisition techniques and issues regarding the low visibility of anatomical structures are addressed by the studies of Yang et al. and Li et al., both focusing on cardiac imaging [14,15]. The former authors applied a joint segmentation method based on a multiview two-task (MVTT) recursive attention model to segment the left atrium of the heart and its scars by mimicking the inspection sequence of radiologists, whereas the latter authors used a deep learning method by developing a multiview recurrent aggregation network (MV-RAN) for the segmentation of each echocardiographic sequence during a full cardiac cycle. Liu et al. also applied an attention model to segment the prostate gland, which exhibits various morphological characteristics [16].

For the segmentation of the lung area, we propose a region growing method where the enclosed rib cage structure is promoted as a peculiarity for a very robust initial demarcation. The specifics of our method along with a brief introduction to the dataset used and the evaluation results follow in later sections. At this point, we review related published works regarding the segmentation of lung regions and ribs, focusing on their strengths and weaknesses and comparing their approaches with ours.

The presence of several insufficiently separated bone structures, some of them irrelevant to the lung area, remains a big concern in many works. Gordienko et al. introduce a deep learning technique for bone shadow exclusion and lung segmentation in 2D CXRs, to assist radiologists in the diagnostic process [17]. However, the introduced additional preprocessing step, which is very complex most of

the time and computationally very demanding, can contribute to the accumulation of errors when used within the context of the overall lung area segmentation. In contrast, in our work we prove that a very simplistic identification of a few rib cage points is enough for quite an efficient overall lung area segmentation algorithm without the intention to preprocess the initial CXR image.

Loog et al. attempted to segment the rib cage by using an iterated contextual single pixel classification approach [18]. The overall relabeling process guarantees that any errors appear in a structured way. They confirmed the challenges of rib cage segmentation, emphasizing the risk of confusion between ribs and clavicles and, besides the poor visibility of the ribs, the potential problem of differences in appearance and size for rib structures localized at the top of the lung fields compared with the rest of the image.

Li et al. qualified ribs overlapping with the lung field as the main noise in lung node detection, pointed out the fact that their shapes are different from each other and subsequently used different templates to describe them [19]. The beforementioned authors introduced a sophisticated filtering process of real rib structures, also employing graph theory. Wessel et al. introduced a unique method addressing both rib segmentation and anatomical labeling in chest radiographs [20]. Based on the idea that the additional information of neighboring ribs leads to a more precise detection of a single rib, they proposed a sequential processing scheme, extending the Mask Region-based Convolutional Neural Network (R-CNN) architecture. As commonly done, Cong et al. considered the segmentation of the two lung fields as a given [21]. In their approach, the generalized Hough transform is employed to localize the lower boundary of the ribs, and based on this, a novel bilateral dynamic programming algorithm delineates the upper and lower boundaries of each rib within the lung area.

Ultimately, our method aims to accurately segment the overall lung regions, benefiting from a very sparse localization of rib cage points, which belong to an imprecise initial contour approximation. In the same manner as our method, Wan Ahmad et al. avoided any training, opting for a straightforward sequential, rule-based algorithm [22]. They highlighted the fact that an unsupervised method is mandatory for a robust Content-Based Medical Image Retrieval System (CBMIRS). Preprocessing deals with different looking X-rays due to different machine configurations, while they managed to estimate the lung border by automatically selecting threshold values for each direction. However, the elimination of clavicles, sides and body artifacts is highly parametric, a fact that increases the risk of low generalization capability. The initial mask is optimized thanks to a clustering approach. Our method can achieve a particularly good quality segmentation result with much fewer parameters since it principally exploits morphological structure information in and around the lungs, which in most cases remains invariable across CXR images.

A notable lung shape deformation appears in radiographs from patients suffering from bacterial pulmonary infections, where the lung boundary might not be determinable even by an experienced observer. This specific application was the focus of Iakovidis et al., who studied CXRs obtained using portable devices with problematic positioning of the patient [23]. They utilized a robust algorithm for the detection of several salient points in the outer rib cage and the spinal cord, which also assisted in a selective thresholding procedure for the different parts. Finally, as done by many other researchers in the field of CXR segmentation, they employed an active shape model (ASM) approach to build a shape prior to their overall algorithm, based on ground truth lung field boundaries. The fact that shape learning is a kernel part in ASM can be a limiting factor in cases where a scarcity of training data that cover a broad range of shape variations is present.

An active shape model (ASM) based on gradient vector flow was applied by Xu et al. [24]. Given a test image, an initial contour estimate is achieved, using a point distribution model. By introducing a global gradient vector flow, the contour is attracted towards the object boundary. Finally, in the method of Annangi et al. [25], the aim is to drive the contour to a given binary prior shape. They emphasized the challenges that are introduced by local minima due to shading effects, by non-rigid shape variations of the lung and by the undesirable presence of unrelated strong edges. In their method, the aim is to

drive the contour to a given binary prior shape. The generation of the initial seed mask is considered very crucial and is completed based on two levels of thresholding.

After the identification of the rib cage points, our algorithm tries to grow the initial segmented lung area towards the actual border of the lung. The border is automatically identified in an unsupervised manner, around the rib cage points. Hence, in comparison to the previously mentioned ASM-based methods, it does not depend on the availability of labeled training data, while it demonstrates a reduced risk of falsely identified border points due to the limited search area.

The available literature, which is reviewed above, is summarized in Tables 1 and 2.

**Table 1.** Summary of approaches for the analysis of lung bone structures found in published literature. The adopted strategy varies and is either the accurate delineation of bone structures for individual analysis or their elimination while segmenting the lung field.

| Algorithm | Main Purpose | Main Methodology | Comments |
|---|---|---|---|
| Gordienko et al. [17] | Assess how clavicles and rib shadows affect lung segmentation | UNet-based convolutional neural network | Improved accuracy by using a preprocessed version of the JSRT dataset without clavicles and rib shadows. Process is sped up by running on a GPU. |
| Loog et al. [18] | Posterior rib segmentation | Iterated contextual pixel classification | Evaluated on the JSRT dataset. Misclassifications appear in a structured way. |
| Li et al. [19] | Rib recognition | Template matching, graph theory and machine learning | Evaluated on the normal X-rays of the JSRT dataset. Overlap with clavicle introduces recognition problems. High sensitivity and specificity. |
| Wessel et al. [20] | Rib segmentation and anatomical labeling | Mask R-CNN | First approach for simultaneous rib detection and segmentation. Improved detection rate. |
| Cong et al. [21] | Eliminate the ribs | Hough transform and dynamic programming | Very high sensitivity and specificity. |

**Table 2.** Summary of related lung segmentation approaches.

| Algorithm | Main Methodology | Datasets | DSC | Ω |
|---|---|---|---|---|
| Wan Ahmad et al. [22] [1] | Oriented Gaussian derivatives filter and Fuzzy C-Means | X-ray datasets from different machine types | - | 0.69–0.87 |
| Iakovidis et al. [23] [2] | Selective thresholding and ASM | Portable chest radiographs of patients with bacterial pulmonary infections | - | 0.91–0.92 |
| Xu et al. [24] [3] | Gradient Vector Flow-based ASM | JSRT and CXR | - | 0.84–0.9 |
| Annangi et al. [25] | Active contours; low-level features at boundary | Shanghai Pulmonary Hospital and other clinical sites in China | 0.88 | - |

[1] Score range from worst to best performing dataset. [2] Average score for left and right lungs. [3] In the paper, scores are given as averages for left and right lungs, for each evaluated dataset. Ω: Jaccard similarity coefficient, DSC: Dice Similarity Coefficient (see Results section).

## 2. Materials and Methods

### 2.1. Proposed Segmentation Method

The proposed method is able to adaptively run per each lung field (right and left) and achieve segmentation in two steps (for the availability of the code see Appendix A). The adaptability of the initially identified region is ensured by automatically selecting rib cage samples, most probably located in the lung area and some support points on the lung border, by incorporating universal a priori

knowledge. The selection of points does not require any previous training, and it will be proven, where necessary, that these points can be indicative of features in challenging segmentation cases.

### 2.2. Lung Region Approximation: A Robust Method

A series of trivial preprocessing steps were taken to reduce irrelevant information as well as to improve the separability of the region of interest. The Contrast Limited Adaptive Histogram Equalization (CLAHE) technique achieves locally significant contrast enhancement. Furthermore, the glenohumeral joint area introduces many challenges during segmentation but its location and size are generally invariable; thus, a straightforward cropping strategy can produce satisfactory elimination results. Afterwards, in order to separate the left and right parts, an automatic symmetry line detection algorithm is employed [26]. This is essentially achieved by comparing features on the image to those in the reflected version of the image. From now on, the segmentation algorithm is applied on each part separately.

The algorithm employs bone structure information inside the lung area. This structural information can help us to better approximate the pixel intensity distribution of the lungs, as described in Figure 1. Based on this distribution, we are able to select a universal upper intensity threshold to initially filter out the background (Figure 2). Moreover, as will be shown in the Results section, abnormal patterns can already be identified by comparing individual CXRs with this distribution.

We found that the Hough Line Transform succeeds in partially identifying the ribs, despite their predominantly curvy shape. This is sufficient for our particular segmentation problem. The two ends of every identified line are included in an initial set of bone points. Afterwards, these points are filtered by reviewing the likelihood of belonging to the rib cage—more specifically in a pattern of almost equally spaced parallel lines (which is usually the case inside a lung field). The third column of Figure 2 shows the distribution of the most probable rib cage points, after the filtering process. Based on their location, it should be straightforward to select only one of the lung contours for further processing (last column of Figure 2). This method is robust, as is also proven during the evaluation process.

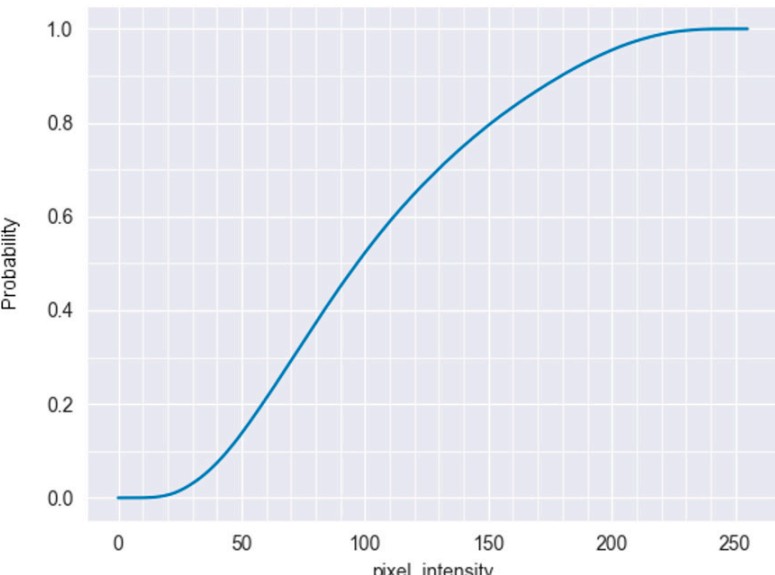

**Figure 1.** Cumulative probability distribution of the intensities of automatically selected most probable bone points inside the lung areas, after aggregating over all Chest X-ray (CXR) images in the dataset. A few points were not taken into account: those around the symmetry line that crosses the sternum, and around the cropped areas around the glenohumeral joints. This plot can be indicative of an appropriate upper intensity limit for an initial thresholding approach. In our specific case, we decided to drop high-intensity values with a total probability of less than 0.25.

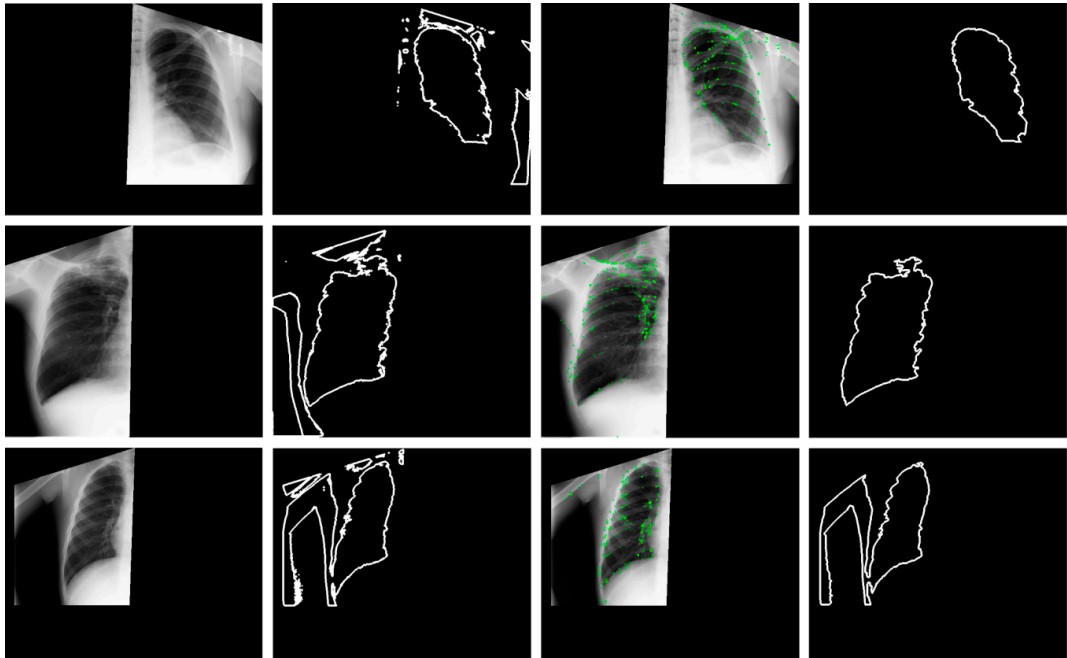

**Figure 2.** Issues after initial segmentation. When the various contours are well separated, the distribution of rib cage points within them can hint at the valid lung contour. For the last row, the algorithm has undesirably identified a region below the arm, connected to the actual lung region. The separation of undesirable areas succeeds, as described in the next section.

An important erroneous case is visible in the last row of Figure 2. Our segmentor identified the biggest part of the actual lung region but failed to separate it from neighboring parts, such as the areas around the glenohumeral joint and the sternum. The method for dealing with this problem is presented in the next subsection.

### 2.3. Minimizing the Irrelevant Lung Area

The solution so far roughly demarcates the actual lung field. However, in many cases the lung contour intensity range was not directly separable from that of other anatomical structures in the image, as the last row of Figure 2 demonstrates. It was experimentally proven that in most cases the separation was possible as a postprocessing step.

In the example in the last row of Figure 2, which is repeated in Figure 3a, there is a clear indication of the separation point between the two contours. We are particularly interested in the points that violate the convexity property of the initial contour, that is, the points where the shape varies with angles greater than 180 degrees. Specifically, first the convex hull of the contour is determined, which is the minimum fitting convex boundary around it. Afterwards, as Figure 3b demonstrates, the convexity defects are isolated. Each of those is represented by the indices of three points in the original contour: the points where the defect starts and ends, and the point that is located farthest away from the two aforementioned points. The distance metric $d$ implemented in OpenCV [27] is calculated as follows: in the 2D image space, given the start and end points $\vec{s}$ and $\vec{e}$ and their integer indices $s, e$ in the contour, the distance of a contour point $\vec{p}$ with index $p \in [s, e]$ is:

$$d = \left| det\left(\vec{d_0}, \vec{d_p}\right) \right| * C \tag{1}$$

with $\vec{d_0} = \vec{e} - \vec{s}$, $\vec{d_p} = \vec{p} - \vec{s}$ and $C = \begin{cases} 0, & \text{if } \vec{d_0} = \vec{0} \\ \frac{1}{\left|\vec{d_0}\right|}, & \text{otherwise} \end{cases}$, a scaling factor.

From now on, we avoid the vector representation of the 2D points, for simplicity in the notation of the mathematical formulas.

Useful prior knowledge for the modification of the contour is the density of the points that probably belong to the rib cage, as presented in Figure 3b. Having these constraints in mind, minor convexity defects, referring to the ones where all three points are very close together, can be fixed by simply filling up their area. A more interesting analysis should be made for the points that very loosely (including their neighborhood) connect two otherwise distinct contours. The way we isolate these points is by evaluating the maximum distance of the farthest point to the start and end points of the convexity defect, as well as the minimum distance of the farthest point to the line that connects the start and end points:

$$\Omega = \left\{ x \mid x \in X,\ s \in S, e \in E \wedge max(d(x,\ s),\ d(x,\ e))\ > T_1\ \wedge d_l(s,e,x) > T_2 \right\} \tag{2}$$

where $X$ denotes the set of farthest points of all convexity defects, $S$ and $E$ correspondingly the sets of starting and ending points of the convexity defects, $d$ the Euclidean distance among two points, $d_l(s, e, x)$ the minimum distance of the point $x$ to the line defined by $s$ and $e$, and $T_1$ and $T_2$ distance thresholds, selected to be sufficiently large (in order to avoid actual lung area separation) relative to the smallest dimension of the bounding box that encloses the identified region up to now.

It must be noted that not all of these points are valid separation points. Following a very conservative approach, we only keep the points in narrow defects. This is done by calculating the angle formed by the lines created from the farthest point and two points on the contour sufficiently close to that point, more specifically one point preceding and one following on the contour. Equation (3) shows this filtering.

$$\Omega' = \{x | x \in \Omega \wedge v_a\ \leq\ T_a\} \tag{3}$$

where $v_a = \sphericalangle C_{i+10} C_i C_{i-10}$, considering C the initial contour points' vector, and $i$ is the index of the examined farthest point on the contour. $T_a$ is the preselected maximum allowed angle.

By now, we are aware of initial contour defects that most probably signal the existence of two unrelated contours. The next step is to achieve this separation. Ideally, the separation should be done by a multitude of curves that represent the complex geometry of the defect, or better said the geometry of the border between the two contours. For simplicity, the separation was made by a line: the one that is connected with the separation point and crosses the defect's bounding box so that the number of background points is maximized, as can be seen in Equation (4) below. Figure 3c shows the separation line selection process.

$$L = \arg\max_{x \in S}\left(\left|\{p \in x : I(p) = 0\}\right|\right) \tag{4}$$

where $S$ is the set of all lines between the defect's farthest point and the points on the line between the defect's start and end, and $p \in x$ denotes the points on the line $x$ represented as pixel coordinates in the image $I$. It must be noted that background pixels in a binary image have a value of 0.

After the separation, it is vital to correctly select the lung field. This was done by utilizing some heuristics, related to the dimensions of the contours and the density of bone points. Figure 3d shows the final selected contour, which confirms that the usage of a straight separation line leads to a suboptimal result. The discovery of the curve that best resembles the border between the two contours would be a very advantageous improvement in the current algorithm.

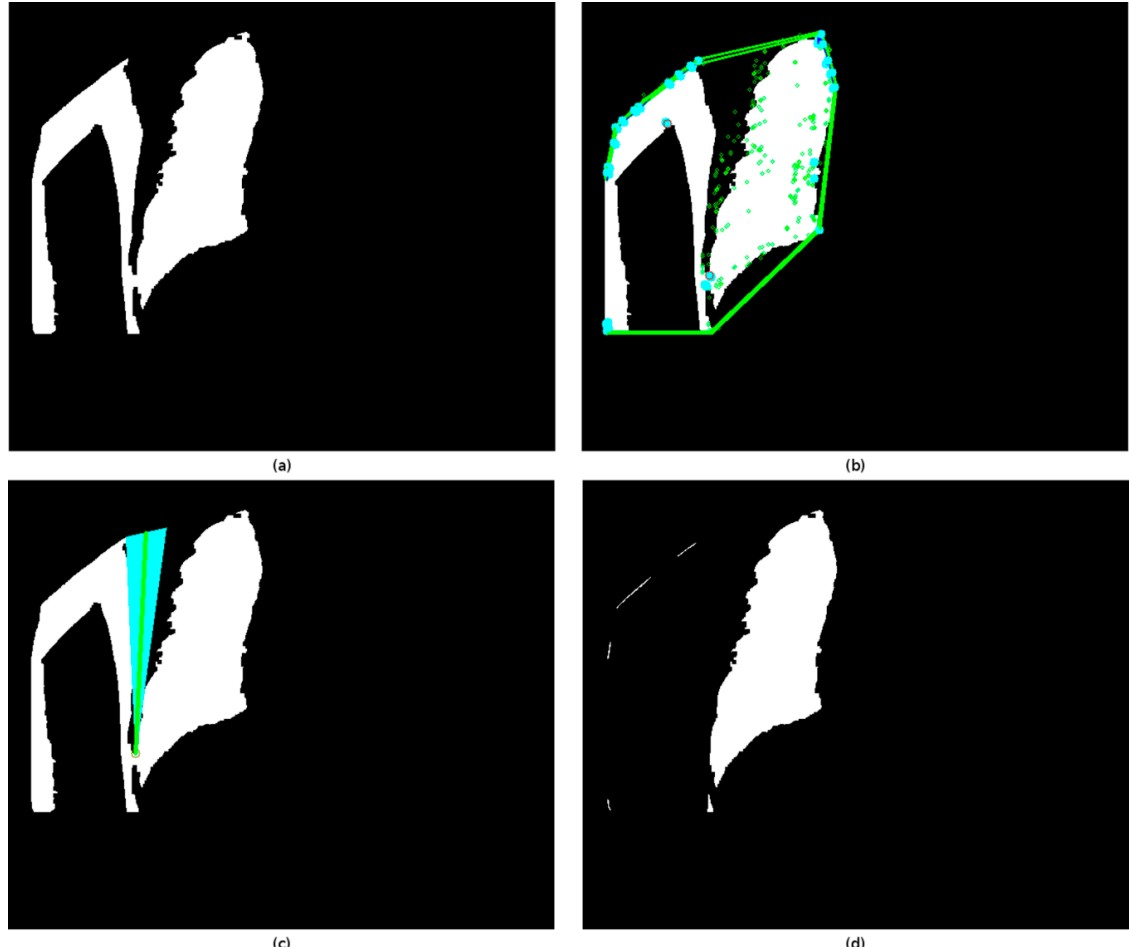

**Figure 3.** Identifying separation points by analyzing convexity defects. (**a**) The initial segmentation result, where two distinct contours are clearly visible. (**b**) Convexity defects: farthest points inside the valleys are illustrated as thick blue circles; start and end points of the defects are connected by a line. Moreover, the rib points are also illustrated as smaller green circles—they will assist in selecting the correct contour after separation. (**c**) A farthest point has been selected as the separation point, and the best candidate separation lines, that cross the most background points, are depicted. These lines start from the farthest point and finish on the line between the start and end points of the relevant defect. The thicker green line crosses the most points inside the defect that do not belong to the contour and qualifies as the best separation line. (**d**) The final contour after separation. Small white residues are eliminated.

## 2.4. Identification of Border Points

Up to now, we were able to estimate the lung area. Unfortunately, there are many abnormalities in its homogeneity. Inevitably, the employed method fails to correctly identify brighter regions that are part of the lung. These parts are usually near the border of the lung area and are very important for several diagnoses.

One can observe that the border of the lung area is (most of the time) clearly separable as it appears as a thin bright line. A sliding window method can be used to filter this line. We are only interested in the line that borders, and is quite close to, the already available approximation of the lung area. Thus, we can limit the scanning to only a small area around the approximation. In our approach, we limit the search further by dropping areas near the vertical center of the image, avoiding the inclusion of parts of the mediastinum.

### 2.5. Stretching the Initial Region

For the same image as the one examined in Figure 3, Figure 4 depicts the result after extending the initial contour to include the border points. The stretching is simply achieved by replacement of the closest continuous segment of the contour with the connected component's points. It is expected that this method is sensitive to falsely detected border lines farther away from the lung area.

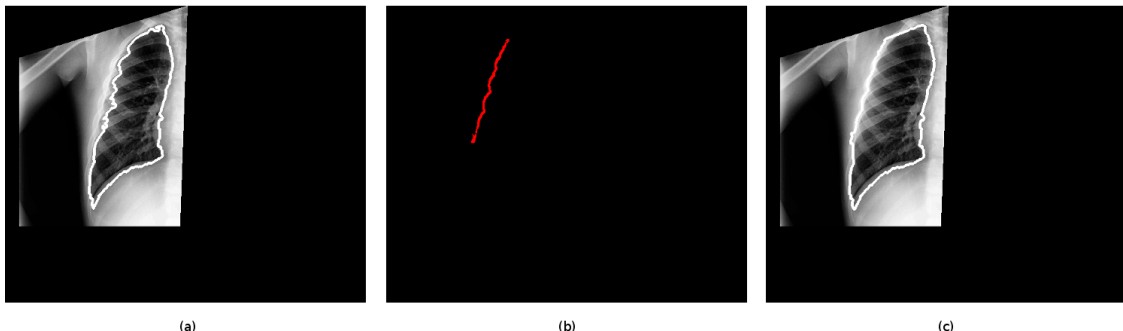

**Figure 4.** The region growing procedure. (**a**) The initial lung contour. (**b**) Identified bright border curve. (**c**) The result after stretching the initial contour to the curve.

### 2.6. Evaluation Dataset

For the evaluation, we utilized the Montgomery County Chest X-ray dataset, publicly available from the National Library of Medicine, National Institutes of Health, Bethesda, MD, USA (see Appendix A) [28,29]. This set contains 138 X-ray images, of which 80 X-rays are normal and 58 are abnormal with manifestations of tuberculosis. A wide range of abnormalities is covered, including effusions and miliary patterns. Many of the diseases can heavily affect the lung shape.

In the vast majority of literature, the segmentation is evaluated on the Japanese Society of Radiological Technology (JSRT) data. This dataset covers a far narrower set of abnormalities. This means that a method developed to segment CXRs of this dataset might not generalize well for many abnormalities. Furthermore, X-rays in this dataset appear smoother intensity-wise due to an older imaging system.

For all these reasons, the Montgomery County dataset was considered representative for the development and evaluation of our method. Manual masks separately for the left and right lung are also provided; these are binary images depicting as white the lung field and as black the background. In addition, although the classification of the abnormalities was out of scope for this work, clinical readings for every CXR are also provided.

For the otherwise straightforward parsing of the images (CXRs and manual masks), it is only worth mentioning that they are read as 8-bit gray-scale images in OpenCV and they are shrunk to $640 \times 525$ pixels. Although this resolution is not that popular in the literature, where in most cases a resolution of $256 \times 256$ is used, it is preferred as a less degraded version of the high-resolution original images. In any case, the results will differ only slightly quality-wise. For modern hardware, the bigger dimensions should not introduce huge delays. Finally, the mapping between the CXR and manual masks can be done immediately, while parsing.

## 3. Results

The validation of the segmentation algorithm aims to compare the detected region with the available ground truth mask in the dataset. To achieve this, three different similarity metrics are employed. Two of them measure the overlapping area and one the distance between the two contours. These measures are widely used in relevant studies [12]. Specifically:

$$\text{Jaccard similarity coefficient: } \frac{|TP|}{|TP| + |FP| + |FN|} \tag{5}$$

$$\text{Dice similarity coefficient: } \frac{2|TP|}{2|TP| + |FN| + |FP|} \tag{6}$$

where |*TP*| is the number of pixels that were correctly classified as lung area, |*FP*| the number of pixels that were incorrectly classified as lung area, and |*FN*| the number of pixels that were incorrectly classified as background.

$$\text{Average Contour Distance: } \frac{1}{2}\left( \frac{\sum_i d(a_i, R)}{|\{a_i\}|} + \frac{\sum_j d(b_j, S)}{\left|\{b_j\}\right|} \right) \tag{7}$$

where *R* denotes the pixels on the boundary of the calculated lung area, and *S* the pixels on the boundary of the ground truth lung area. Therefore, we average the minimum distances between pixels of one boundary to the other and the reverse, and then we average the two results.

After executing the algorithm on the full dataset, we obtained an average Jaccard similarity coefficient of 0.862, Dice similarity coefficient of 0.923 and Average Contour Distance of 7.4 pixels. From this point, one can perform an analysis at the image level. The algorithm demonstrates a Jaccard score above 0.84 and close to that of the state-of-the-art methods for about 77.5% of the images in the dataset. On the other hand, the segmentation of about 5.8% of the images is very questionable, where the Jaccard score was between 0.59 and 0.77. Table 3 compares the results of our method, its strengths and its weaknesses to other recent studies, which apply neural networks for the segmentation of lung fields.

Table 4 summarizes an analysis of misdetections, where only cases of CXRs with Jaccard score below 0.8 are considered. It is understandable that the intensity homogeneity of the pixels inside the lung area affects the algorithm's segmentation capability. Nevertheless, as the table shows, some of these cases are of crucial medical interest. In a further analysis, we prove that the probability distribution of intensity values inside the lung area can assist in identifying such pathological patterns. For example, Figure 5 demonstrates that an abnormality in the left lung (pleural effusion in this case) is confidently recognized, using the cumulative probability of intensities inside the lung area, sampled around the most probable rib cage points, shown in the left-most image. Cases where the field of one or both lungs extends to bright regions can be assigned a lower segmentation confidence value by the algorithm.

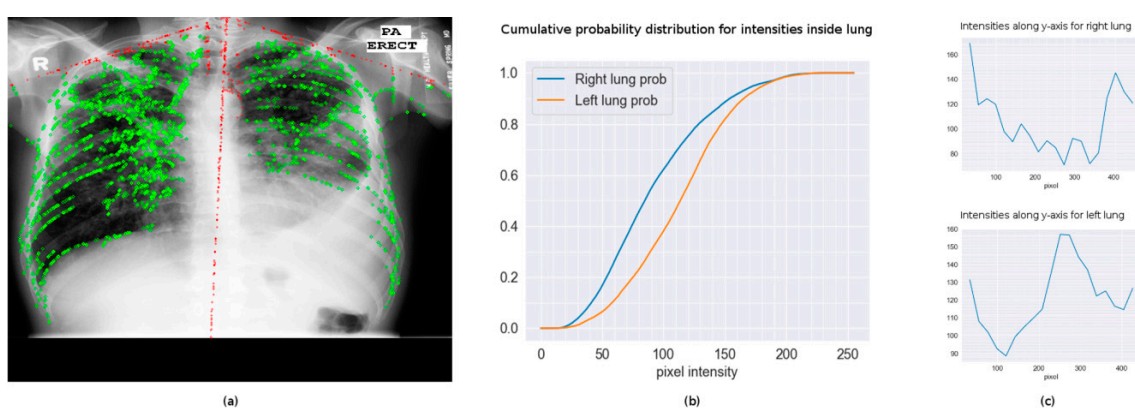

**Figure 5.** Recognition of pleural effusion in the left lung. (**a**) Automatically selected most probable lung points used as samples for the characterization of the lung area. The points around the symmetry line at the center and around the preselected shoulder area have been excluded as they most probably belong to irrelevant structures (these points are illustrated thinner with red color). (**b**) Cumulative probability distribution of intensities inside the lung area calculated around the sample points. (**c**) The intensity along the *y*-axis of the image for the right lung (upper) and for the left lung (lower), top to bottom, averaged along the corresponding samples on the *x*-axis.

**Table 3.** Comparison of our method with very recent neural network approaches.

| Authors (Year) | Main Method | Dataset | Jaccard | Dice | Strengths | Weaknesses |
|---|---|---|---|---|---|---|
| Kalinovsky et al. (2016) [30] | Encoder/Decoder CNN | Tuberculosis portal and JSRT | - | 0.962 | Uniform Deep Learning approach. | Hardware-demanding training. |
| Novikov et al. (2018) [31] | InvertedNet with Exponential Linear Units | JSRT | 0.95 | 0.974 | Copes with overfitting and imbalanced data. Reduces parameters. Segmentation of lungs, clavicles and heart. | Training and testing on the same dataset. Computational feasibility trade-off. |
| Arbabshirani et al. (2017) [32] | Registration-based and patch-based CNN | Geisinger and JSRT | - | 0.88–0.96 | Heterogeneous dataset. Multiscale network evaluated. | Hardware-demanding training. Coarse lung boundaries in some images. |
| Souza et al. (2019) [33] | Patch-based AlexNet, ResNet-18 with 2 deep CNNs | Montgomery County | - | 0.94 | Second CNN for more complex cases. Better segmentation of lungs with dense abnormalities. | Postprocessing required after the first network. Resizing of images required due to hardware limitations. Second network does not ensure quantitative improvement and leads to decreased performance. Many parameters. |
| Dai et al. (2018) [34] | Structure-Correcting Adversarial Network (SCAN) | JSRT, Montgomery County | - | 0.973 | Segments lung fields and heart. Limited training data. Generalizes to different patient populations and disease profiles. | Like many other methods, labeled data are a necessity. |
| Oh et al. (2020) [35] | Patch-based (FC) DenseNet103 | Mixture of public CXR datasets | 0.932–0.955 | | Few trainable parameters. Provides clinically interpretable saliency maps, which are useful for COVID-19 diagnosis and patient triage. Patch training leads to smaller network complexity and augmentation of dataset. Performs classification. | |
| Huynh et al. (2019) [36] | Hybrid Network with network individuals | Hoan My Hospital | - | 0.87 | Huge improvement compared to applying a traditional CNN to the same dataset. Addresses the challenge of segmenting large-size chest X-ray images. | Not evaluated on a standard dataset. Small testing set. Boundaries not that smooth. FPs or FNs when similar density between regions or high-curvature lung regions. |

**Table 3.** *Cont.*

| Authors (Year) | Main Method | Dataset | Jaccard | Dice | Strengths | Weaknesses |
|---|---|---|---|---|---|---|
| Chen B et al. (2020) [37] | Two-Stream Collaborative Network (TSCN) with U-net at segmentation stage | JSRT, Montgomery County and NIH | - | 0.973 | Performs classification. Few training images. Combined datasets for training and validation. | Poor performance with the Infiltration group. |
| Chen HJ et al. (2020) [38] | CNN-based architectures applied on binarized images | Montgomery County and private clinic in India | 0.842 | 0.893 | Fast training, low storage requirements. Contrast enhancement helps a lot to improve Dice score. | Contrast enhancement usefulness in terms of Jaccard measurement improvement depends on the selected Network architecture. Relatively small validation and testing set. |
| Our method | Rib cage points-driven region growing | Montgomery County | 0.862 | 0.923 | Straightforward unsupervised method. Can be used for rapid triage of patients. | Depends on at least some visibility of the rib cage and a distinguishable border curve. A few parameters still have to be selected by intuition. |

CNN: Convolutional Neural Network, JSRT: Japanese Society of Radiological Technology, NIH: National Institutes of Health.

**Table 4.** Segmentation failure scenarios. There might be multiple failure scenarios for each CXR; this is why the location column indicates the specific part of the lungs' field where a failure is observed. This analysis is only done for CXRs with low evaluation scores. TB: tuberculosis.

| Count of Failed Cases | Location | Reason/Diagnostic Relevance |
|---|---|---|
| 5 | Bottom part | Bright region probably due to pleural effusion |
| 4 | Bottom part | Pleural fluid causes bright areas |
| 3 | Multiple | Infiltrates due to TB—brighter regions, bone structures not sufficiently visible |
| 1 | Inner part of right lung | Increased length of cardiac silhouette, probably due to pericarditis |
| 1 | Left lung's outer side brighter | Bright area likely because of infiltrate due to pneumonia |
| 1 | Bottom part | Bright region, localized pleural peel |
| 1 | Inner part of right lung (right hilum) | Congestive heart failure or infiltrate due to TB |
| 1 | Inner part of left lung | Extended bright region due to cardiomegaly |
| 2 | Top-left/top-right part of right lung | Inaccurate symmetry detection algorithm crop or glenohumeral joint area crop. Diagnosis irrelevant. |
| 7 | Multiple | CXRs either normal or pathological. The failure is irrelevant to the diagnosis and is probably due to poor contrast. |

The results show that this method, despite its simplicity, gives very accurate results for the vast majority of images. Furthermore, it is shown that even a very vague recognition of rib cage points can guide the overall lung area segmentation. For algorithm output, graphs and images see Appendix A.

## 4. Discussion

In this study, we achieved the delineation of the lung fields in CXR images, based on the density field of automatically detected lung area bone points. We proved the feasibility of a robust and straightforward region growing procedure from an initial segmentation around these points up to a set of support points identified on top of the actual lung border. In a dataset that contains a great variety of shapes and abnormal patterns, our method achieves a very good performance, which is directly comparable to those of recent algorithms that often employ Convolutional Neural Networks (Table 3). Finally, we demonstrate, through a probabilistic analysis, that the bone points can be used as a strong lung area feature, even in challenging cases with huge lung area intensity inhomogeneity, such as those presenting an extended pleural effusion or other chest pathologies. In the scenario of a high-volume radiology practice, such as a hospital emergency department, the presented algorithm could be used as a rapid triage tool by flagging CXRs with disconcordant cumulative intensity distributions as "failed", given the fact that failed cases actually allude to various chest pathologies that require further interpretation by an expert radiologist. Our method includes specific limitations already mentioned in Table 3. It requires a minimum level of visibility of the rib cage and a distinguishable border curve. The beforementioned parameters are prone to variations that depend on the CXR acquisition technique, specific pathologies and the patient's body type. Moreover, some parameters of our algorithm, as already mentioned, have to be selected by intuition such as the edge detection thresholds of the Hough Line Transform procedure. As a future step, we aim to apply and evaluate our algorithm on a mixed dataset of COVID-19 CXRs (such as the "COVID-19 Image Data Collection" by Cohen et al., which is a work in progress [39]) and other types of pneumonia (viral and bacterial, including tuberculosis), and to modify it in order to correctly label cases.

**Author Contributions:** Conceptualization, V.B. and N.S.I.; Data curation, V.B. and N.S.I.; Formal analysis, V.B. and N.S.I.; Methodology, V.B. and N.S.I.; Project administration, N.S.I.; Software, V.B.; Validation, V.B. and N.S.I.; Writing—original draft, V.B. and N.S.I.; Writing—review and editing, V.B. and N.S.I. Both authors agree to be accountable for all aspects of the work in ensuring that questions related to the accuracy or integrity of any part of the work are appropriately investigated and resolved. All authors have read and agreed to the published version of the manuscript.

**Funding:** This research received no external funding.

**Acknowledgments:** We would like to thank Konstantinos (Kostas) Sechidis, machine learning researcher and honorary research fellow at the University of Manchester—School of Computer Science, for proofreading the manuscript and providing invaluable comments and recommendations.

**Conflicts of Interest:** The authors declare no conflict of interest.

## Appendix A

- Availability of data and material: Algorithm output (images and graphs) is available at: https://drive.google.com/drive/folders/1iaq4mFhgM2Loedlj_bXBKp-ZeIPVCwEF
- The evaluation dataset used is publicly available at: https://lhncbc.nlm.nih.gov/publication/pub9931
- Code availability: https://bitbucket.org/vmposdel/cxr_image_segmentation/

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
