# Peer review of "Lung Field Segmentation in Chest X-rays: A Deformation-Tolerant Procedure Based on the Approximation of Rib Cage Seed Points"

_applsci, doi:10.3390/app10186264_

Round 1

Reviewer 1 Report

The manuscript by Bosdelekidis et. al. describes the delineation of the lung fields in CXR images based on the density field of automatically detected lung area bone points. They proved the feasibility of a region-growing procedure from an initial segmentation around these points up to a set of support points identified on top of the actual lung border. The results demonstrated good performance for a dataset that contains a great variety of shapes and abnormal patterns. They also demonstrated through a probabilistic analysis that the bone points can be used as a strong lung area feature, even in challenging cases with huge lung area intensity inhomogeneity. The paper is presented in a logical and clear manner, but it needs improvements before it can be considered for publication. Here are my concerns and I would appreciate it if the authors can further clarify on or improve these issues.

  1. In the Introduction part, it will be better if the authors can compare the pros and cons of different methods with their proposed one in the paper, instead of just listing what other people did and their conclusions.
  2. In Line 194, why did the author choose the farthest point in Figure 3c? For example, for another dataset, how should the authors decide which farthest point should be chosen? Does this choice have any effects on the final segmentation results?
  3. In Line 211, how did the author determine the absolute values of thresholds? Can the authors provide some quantitative explanations?
  4. In line 225, I am sorry but I am still confused about how the separation line in Figure 3c was selected. Basically, how did the authors get the blue triangle in Figure 3c? Based on Equation (1) and (2)? Can the authors provide more details or explanations?
  5. In the Discussion part, it’s more like a Conclusion part, can the authors discuss some limitations or future directions of this proposed algorithm? 

Author Response

Thank you for your consideration and fruitful comments. We reply to them point by point below:

  1. We added points of comparison regarding already published approaches to ours. See lines 93-100, 103-106, 124-129, 137-139, 148-152 of the revised manuscript.
  2. The selection of the farthest point is done as in Equation 1, line 241 in the revised text. This point indicates the best position on the contour (from all candidates around the convexity defect) to do the separation (if any) and this standardized selection should give the optimal results.
  3. An explanation is added in lines 268-272 in the revised manuscript.
  4. Only the selection of the separation line was explained in the initially submitted manuscript but now we added Equation 4 in line 288. Figure 3c depicts the best candidate separation lines (crossing the most background points) in blue color, which are basically lines starting from the separation point and finishing on the line between the start and end points of the defect. This additional info which was indeed missing in now appended in lines 251-253.
  5. In the Discussion section we have added text regarding the limitations and future directions of our work. See lines 432-440 of the revised manuscript.

Reviewer 2 Report

Interesting work for lung segmentation from CXR and could be well adaptive for COVID patients as well. Only a couple of suggestions to improve the work:

1. There are some recently published work for CXR and Chest CT and in general medical image segmentation, which are very relevant, that the authors need to refer to:

Hu, Shaoping, et al. "Weakly supervised deep learning for covid-19 infection detection and classification from ct images." IEEE Access 8 (2020): 118869-118883.

Yang, Guang, et al. "Simultaneous left atrium anatomy and scar segmentations via deep learning in multiview information with attention." Future Generation Computer Systems 107 (2020): 215-228.

Li, Ming, et al. "MV-RAN: Multiview recurrent aggregation network for echocardiographic sequences segmentation and full cardiac cycle analysis." Computers in Biology and Medicine (2020): 103728.

Liu, Yongkai, et al. "Automatic prostate zonal segmentation using fully convolutional network with feature pyramid attention." IEEE Access 7 (2019): 163626-163632.

2. The quantitative results, e.g., Dice scores etc., may need to be highlighted in the paper.

3. How efficiency is the proposed method?

4. Add a discussion section that discuss the limitations of the current study especially compared to recently published deep learning based methods.

Author Response

Thank you for your consideration and fruitful comments. We reply to each one of them point by point below. Please note that every change is highlighted with yellow colour in the revised manuscript.

  1. We added and briefly analyzed the recommended published work in the Introduction section. See lines 68-85 of the revised manuscript.
  2. We highlighted the results as requested (see lines 355-356) and also included them separately in Table 3 which compares our results to other CNN approaches.
  3. Regarding the efficiency of our method we commented on it in the “Strengths” column of Table 3.
  4. We added text regarding the limitations of our method in the Discussion section (see lines 432-436). Moreover, in Table 3, we provide a direct and thorough comparison of our method’s results, strengths and weaknesses against published CNN approaches.

Round 2

Reviewer 1 Report

The authors have addressed the comments

Author Response

Thank you for your comments.

Regarding the suggestion of a minor spell check, the manuscript has been reviewed and corrected by a researcher with full working proficiency in English, who is mentioned in the Acknowledgements section. Any minor changes are highlighted in yellow.